# Pancreatic Diseases and Microbiota: A Literature Review and Future Perspectives

**DOI:** 10.3390/jcm9113535

**Published:** 2020-11-01

**Authors:** Marcantonio Gesualdo, Felice Rizzi, Silvia Bonetto, Stefano Rizza, Federico Cravero, Giorgio Maria Saracco, Claudio Giovanni De Angelis

**Affiliations:** Gastroenterology and Digestive Endoscopy Unit, AOU Città della Salute e della Scienza, University of Turin, 10126 Turin, Italy; marcantonio.gesualdo@gmail.com (M.G.); silvia.bonetto@edu.unito.it (S.B.); stefanorizza@live.it (S.R.); fed.cravero@gmail.com (F.C.); giorgiomaria.saracco@unito.it (G.M.S.)

**Keywords:** pancreatic diseases, microbiota, microbiome, gut microbiota, acute pancreatitis, chronic pancreatitis, diabetes mellitus, pancreatic ductal adenocarcinoma, pancreatic cystic neoplasms

## Abstract

Gut microbiota represent an interesting worldwide research area. Several studies confirm that microbiota has a key role in human diseases, both intestinal (such as inflammatory bowel disease, celiac disease, intestinal infectious diseases, irritable bowel syndrome) and extra intestinal disorders (such as autism, multiple sclerosis, rheumatologic diseases). Nowadays, it is possible to manipulate microbiota by administering prebiotics, probiotics or synbiotics, through fecal microbiota transplantation in selected cases. In this scenario, pancreatic disorders might be influenced by gut microbiota and this relationship could be an innovative and inspiring field of research. However, data are still scarce and controversial. Microbiota manipulation could represent an important therapeutic strategy in the pancreatic diseases, in addition to standard therapies. In this review, we analyze current knowledge about correlation between gut microbiota and pancreatic diseases, by discussing on the one hand existing data and on the other hand future possible perspectives.

## 1. Introduction

The human gastrointestinal (GI) tract is colonized by a rich microbial community consisting of more than 10^14^ microorganisms, defining microbiota, and more than 5,000,000 genes defining microbiome [1,2]. The microbiota is composed of bacteria, viruses and yeasts [3]. In healthy conditions, these microorganisms colonize mucosal surfaces, particularly the large intestine, and talk closely with them; in this way they regulate important physiological functions [4,5]. First, they are involved in metabolism of nutrients and drugs and vitamin production [6]. Then, through food fermentation, bacteria produce some short-chain fatty acids (SCFAs), for example butyrate, which have trophic effects on the GI epithelium [7]. Furthermore, gut microbiota influences the immune system through its antigenic effects. The interaction between gut microbiota, intestinal epithelial cells and the mucosal immune system creates an environment that prevents overgrowth of the host pathogenic microorganisms [8] and limits the colonization of the intestinal tract by foreign pathogens [9,10,11]. 

In healthy people gut microbiota is characterized by richness in microorganisms and high diversity of species. This situation is called *eubiosis*. In this microenvironment, bacteria are predominant and represent the main group of microorganisms that are strictly anaerobics and extremophiles. *Firmicutes* and *Bacteroidetes* represent the main bacterial phyla, up to 85–90% of total microorganisms, while *Actinobacteria* and *Proteobacteria* are less plentiful, representing up to 10% [12].

In this condition, commensal bacterial species are predominant compared to pathological ones. 

Conversely, when this ecosystem balance is perturbed (i.e., by use of antibiotics, motility disorders, diet, host genetic features, etc.) [3], there is a condition called *dysbiosis,* characterized by a lowering in diversity of bacterial species, with abundance of pathogenic ones, and a loss of microbiome physiological functions [13,14]. In some cases, in this dysbiotic environment, there is a reduction of tight junctions between enterocytes, leading to a compromised function of mucosal barrier integrity; this alteration, named leaky gut, sometimes allows bacterial translocation and plays a key role in the development of GI and systemic diseases [11,15]. 

The composition of gut microbiota may be strongly influenced by both pathological conditions and environmental factors, such as age, diet, drugs, stress [16]. Besides, the abundance and the variety of different species within an individual microbial system (i.e., a single sample) is called α-diversity, while β-diversity refers to differences between microbial communities from different environments (i.e., different samples or different individuals) [17].

In clinical practice, we may manipulate microbiota by administering prebiotics, probiotics or synbiotics, through fecal microbiota transplantation (FMT). Prebiotics are defined as “a substrate that is selectively utilized by host microorganisms conferring a health benefit” [18]; the main prebiotics that have healthy benefits are non-digestible fructooligosaccharides (FOS) and galactans (GOS), preferentially metabolized by *Bifidobacterium* spp. Other examples of prebiotics are polyunsaturated fatty acids (PUFAs) and inulin [19]. Intestinal microorganisms can readily utilize prebiotics, transforming them in metabolic products, such as SCFAs, i.e., propionate, butyrate, acetate. These products are crucial for correct intestinal health. Prebiotics are now largely used in clinical practice for treating many diseases, such as inflammatory bowel disease (IBD) [20], irritable bowel syndrome (IBS) [21], metabolic syndrome [22]. Conversely, probiotics are defined as “live microorganisms that confer a health benefit on the host” [23]. Probiotic foods contain safe live microbes with sufficient evidence for a general beneficial effect in mammals [24]. Synbiotics are a mixed product with a combination of probiotics and prebiotics. Finally, FMT consists of “the infusion of faecal samples from a healthy donor to the GI tract of a recipient patient, in order to cure a specific disease, improving alteration of gut microbiota” [25]. To date, the only indication to perform FMT is the recurrent and refractory (non-responder to conventional antibiotics, i.e., vancomycin, fidaxomicin or metronidazole) *Clostridium difficile* infection with an efficiency rate standing at more than 80–85% [25]. 

Due to these novelties, microbiota is now a worldwide field of interest and investigations are growing in the recent years. Several studies analyzed the involvement of intestinal dysbiosis in the development of intestinal and extra-intestinal diseases, such as IBD [26], celiac disease [27], IBS [28], multiple sclerosis [29], rheumatologic diseases [30], Alzheimer’s disease [31], colorectal and gastric cancer [32]. On the contrary, data about correlation between microbiota and pancreatic diseases are still scarce and controversial.

Few studies described the presence of bacteria in pancreatic tissue; they found bacteria in pancreatic ducts of subjects with chronic pancreatitis or in pancreatic tissue of pancreatic cancer patients. Instead, recently, some authors analyzed microbiome in pancreatic samples and duodenal tissues from patients underwent pancreatectomy, finding a similar bacterial DNA profiles; this may suggest a bacterial translocation from the gut into the pancreas [33]. Due to the impossibility to collect pancreatic tissues routinely, a lot of studies analyzed gut microbiome from fecal samples.

In this review, we analyze the actual available data in literature about microbiota and pancreas in health and disease. 

## 2. Methods

A literature search was performed in PubMed, Scopus, Web of Science, Cochrane Library databases and Embase. The search included papers published from 1 January 1993 to 1 March 2020. Only English studies were considered. All authors participated in the search process and in the critical analysis of selected publications. Keywords used were: “microbiota”, “pancreas”, “acute pancreatitis”, “chronic pancreatitis”, “pancreatic cystic neoplasms”, “pancreatic ductal adenocarcinoma”, “diabetes mellitus”, “neuroendocrine tumors”, “probiotic”, “prebiotic”, “synbiotic”. 

## 3. Microbiota in Healthy Pancreas

Recently, the pancreatic physiological functions have been shown to have an impact on intestinal microbiota and vice versa [34], by a cross-talking system, in health and disease; hence, it could be possible to talk about “microbiota-pancreas axis” [35].

In an important study, Sun et al. [36] demonstrated that pancreatic β-cells, in mice, produced cathelicidin-related antimicrobial peptide (CRAMP), a protein with antimicrobial activity through bacterial membrane permeabilization [37,38]. They noticed that pancreatic CRAMP expression was induced by SCFAs derived from the gut microbiota, underling the cross-talking system. In a recent study, Ahuja et al. [39] used a mice model with pancreatic acinar cell–specific deletion of Orai1, a Ca^2+^ channel necessary for exocytosis of pancreatic antimicrobials. They found that Orai1-deficient mice showed an altered intestinal microbiota with bacterial overgrowth. In particular, *Proteobacteria* were two-fold increased, including increases in *Succinivibrionaceae* and *Enterobacteriaceae*, and *Prevotella* spp. The authors did not find differences on expression of intestinal antimicrobials in Orai1-deficient mice, proving the role of the pancreas in control of intestinal microbiota. Moreover, Orai1-deficient mice developed spontaneous intestinal inflammation characterized by intestinal CD3+ T-cell infiltration and died after three weeks. This data could be translated on humans, because patients that carry ORAI1 nonsense mutations have susceptibility to GI infection and diarrhea [40]. In an American cohort of patients who underwent a pancreaticoduodenectomy, Rogers et al., studied fecal microbiota composition compared to a group of healthy individuals. Interestingly, surgical patients exhibited fecal dysbiosis and showed an increase in *Klebsiella* and *Bacteroides*, while anaerobic taxa (i.e., *Faecalibacterium prausnitzii* and *Roseburia* species) decreased [41]. After DCP there are several anatomical alterations apart from the pancreatectomy (i.e., partial gastrectomy, the biliary-digestive anastomosis). In this scenario, also the gastric juice changes its composition. These data suggest that both pancreatic and gastric juice, with their antimicrobial activity, are able to modify gut microbiome.

## 4. Microbiota in Pancreatic Diseases

### 4.1. Microbiota and Acute Pancreatitis

Acute pancreatitis is caused by gallstones or acute alcoholic intake in the majority of cases. Rare causes of pancreatitis are hypertriglyceridemia, hypercalcemia, genetic causes, smoking, viruses such as cytomegalovirus or herpes simplex virus [42]. The main therapy for acute pancreatitis is early fluid resuscitation; the latter plays a key role by improving the tissue oxygenation and the microcirculation perfusion in order to preserve on the one hand pancreatic function (avoiding further pancreatic necrosis), and on the other hand, renal, cardiac and intestinal perfusion [43,44]. The use of antibiotics is not recommended routinely in severe acute pancreatitis (SAP), but only in the presence of infection, as cholangitis associated to biliary pancreatitis, and suspicion of pancreatic necrosis. In AISP (the Italian Association for the Study of the Pancreas) guidelines [45], the routine use of probiotics is not recommended. Besides, the use of probiotics in clinical practice is not cited in the recent American guidelines addressing the management of pancreatic necrosis [42]. 

Acute pancreatitis determines hypovolemia that induces splanchnic vasoconstriction to preserve vital organs. Sometimes, this situation favors leaky gut. Indeed, high intestinal permeability promotes bacterial translocation and endotoxemia, with increased risk of infection of necrotic tissues [46]. Moreover, small intestinal bacterial overgrowth (SIBO) and bacterial translocation may be responsible for the majority of the pancreatic complications, but evidences are still scarce. Capurso et al. [44] showed that intestinal ischemia damages gut barrier with bacterial translocation, endotoxemia and rarely multi-organ dysfunction syndrome. 

Tan et al. analyzed differences in microbiome in patients with mild acute pancreatitis and SAP and a control group [47] (Table 1). They found fecal dysbiosis characterized by an increase in *Enterococcus* species and a decrease in *Bifidobacterium* species in SAP, when compared with other patients with mild form and control group. These data were in line with other studies [48]. 

In a recent case control study, Zhang et al. [49] found that fecal samples from 45 patients with acute pancreatitis contained fewer *Actinobacteria* and *Firmicutes* and more *Bacteroidetes* and *Proteobacteria* than those from 44 healthy controls. 

The first randomized controlled trial (RCT), assessing the usefulness of probiotics on clinical outcomes of acute pancreatitis (AP), was conducted by Olah et al., in 2002 in Hungary [50] (Table 2). 

The results were encouraging. Forty-five patients were recruited, 22 of them received *Lactobacillus plantarum* (*LP*) 299 with a substrate of oat fiber for 1 week by nasojejunal tube; instead, 23 patients were treated with a similar product with *Lactobacillus* inactivated by heat. Only one patient in the first group had pancreatic necrosis versus seven patients in the control group. Similar data was published by Qin et al. [48]. The authors enrolled 74 patients with AP and divided them into two groups: the first was treated with classical parenteral nutrition and the other was treated with enteral feeding and probiotics, containing living *LP*, administered through a nasojejunal tube. Analyzing fecal samples, they found that the second group was colonized with lower number of pathogenic organisms than the first one; besides, the first group had increased permeability with an increase of *Enteroccoccus* spp. and a reduction in *Lactobacteria* and *Bifidobacteria*. Thus, *LP* enteral feeding induced beneficial effects on maintaining the integrity of intestinal mucosal barrier and reducing the risk of septic complications. Olah et al., in 2007, published another RCT [51] with a new synbiotic composition, named “Synbiotic 2000”, in the treatment of patients with AP. Sixty-two patients were randomized to probiotics plus prebiotics treatments or to prebiotics treatment alone. Patients in the first group had less complications (i.e., systemic inflammatory response, multiorgan failure) than the control group, but the difference was not statistically significant. In the multi-center, nationwide, double blind placebo-controlled clinical trial called PROPATRIA study [52], 298 patients were enrolled. The probiotic group was treated with *Ecologic 641*, administered twice daily by nasojejunal feeding. It consisted of six strains of microorganisms (four types of *Lactobacilli* and two of *Bifidobacteria*). Instead, the control group received placebo. The authors found [53] that there was no difference in the risk of infections between patients treated with probiotics and the control group (30% versus 28%). However, the data were dramatically different concerning mortality. Indeed, 24 of 152 patients treated with probiotics (16%) died versus 9 of 144 (6%) in the placebo group. In particular, a high risk of mesenteric ischemia was seen in the probiotic group versus the control group. However, Bongaerts et al., made a reassessment of this study eight years later, analyzing the weak points of PROPATRIA trial [54]. One of them was the latency time in the first administration of probiotics; indeed, some patients were treated 24 h after onset of symptoms. Furthermore, there were errors in randomization; in fact, onset of multi-organ failure was already present during admission in more patients in the first group than in the placebo group (41 patients versus 23 patients). 

Finally, they suggested, that future studies should evaluate the usefulness of high dose of probiotics, but caution is mandatory to prevent bacterial overgrowth. 

Gou et al., in a meta-analysis published in 2014 [55], asserted that probiotics showed neither adverse nor beneficial effects on the clinical conditions of patients with SAP. However, they highlighted a significant heterogeneity in the studies of clinical outcomes. However, they highlighted a significant heterogeneity in the studies of clinical outcomes. 

Currently, data are not sufficient to draw a conclusion regarding the effects of probiotics on patients with SAP, since they are controversial and heterogeneous. Future studies should improve the study designs, for example, detecting a peculiar strain of microorganisms (i.e., the type of probiotic), the correct dose of probiotics, or by a standardization of treatment duration.

### 4.2. Microbiota and Chronic Pancreatitis

Chronic pancreatitis (CP) is a persistent disease characterized by chronic inflammation of the gland with typical findings, such as Wirsung dilatation, intraparenchymal calcifications, gland atrophy, and development of fibrosis [56]. Chronic pancreatitis leads to progressive endocrine and exocrine dysfunction. Data suggest that patients with CP display dysbiosis with growth of pathogenic bacteria. This situation is related to the reduction of antimicrobial peptides in the pancreatic juice [57]. Capurso et al. analyzed by a systematic review and meta-analysis the relationship between SIBO and CP [58]. The authors proved that about 36% (range between 14–92%) of CP patients are affected by SIBO, detected by a glucose or lactulose hydrogen breath test. However, because of high heterogeneity in the results, the relationship is unclear, and further research is needed.

Pancreatic enzyme replacement therapy (PERT) is able to alleviate exocrine pancreatic insufficiency (PEI) related symptoms, helping fat absorption and improving nutritional status of patients [59]. On the other hand, some authors hypothesize that PERT could directly modify gut microbiota, but the mechanism is still unknown. In the study by Nishiyama et al. [60], the authors evaluated the gut microbiota from fecal samples in mice treated with PERT versus controls. Their findings supported the hypothesis that PERT alleviates PEI-associated symptoms by ameliorating digestive activity and by changing the composition of gut microbiota. In particular they found a relative abundance of *Akkermansia muciniphila* and *Latobacillus reuteri* in mice treated with PERT.

In fact, researchers found a relative abundance of two types of microorganisms in fecal samples from pancrelipase-treated mice compared with controls, i.e., *Akkermansia muciniphila* and *Lactobacillus reuteri*. In particular, the first one, found 58-fold higher in the PERT group, is known to enhance intestinal barrier function by promoting mucus thickness and tight junction protein expression reducing the leaky gut [61]; so, this study suggests that PERT may help to maintain *eubiosis*.

Recent studies highlighted the involvement of immune responses against intestinal microbiota in the development of CP [62]. For example, the activation of pattern recognition receptors (PRRs), i.e., toll-like receptors (TLR) and nucleotide-binding oligomerization-like (NOD) receptors that detect pathogen-associated molecular patterns (PAMPs) derived from the intestinal microbiota have been reported to play a critical role in the development of experimental CP [63].

Proton-pump inhibitors (PPI) are not recommended in patients with CP, but in refractory steatorrhea they could be helpful. In fact, patients with severe forms of inflammation can have lower bicarbonate secretion that cannot neutralize the acidity in the duodenum [64].

In a randomized, controlled, double-blinded trial, dos Santos et al., used synbiotics for changing intestinal microenvironment of patients with CP [65]. Synbiotics was composed of 12 g/day of *Lactobacillus casei*, *Lactobacillus rhamnosus*, *Lactobacillus acidophilus*, *Bifidobacterium bifidum* and FOS, and was administered to the intervention group; instead, 12 g/day of medium absorption complex carbohydrate was administered to the control group. The authors found that synbiotics improved clinical and laboratory outcomes in patients with CP. Dylag et al. [66] found that synbiotics may help restoration of the gut microbiota through the fermentation of FOS and the consequent increase in *Bifidobacteria* and the reduction in pathogenic bacteria. The use of prebiotics and synbiotics in patients with CP promotes a reduction in intestinal stool frequency [65], but their clinical relevance role warrants additional investigation.

Collectively, these promising results suggest that manipulation of microbiota, due to low cost and high manageability, may represent a new therapeutic frontier in CP.

### 4.3. Microbiota and Autoimmune Pancreatitis

Autoimmune pancreatitis (AIP), described for the first time in 1995, is a rare fibro-inflammatory form of CP with typical imaging and histological findings, caused by autoimmune abnormality [67,68]. There are two subtypes of AIP: type 1 and type 2. Type 1 is the pancreatic manifestation of a systemic Immunoglobulin G4–related disease (IgG4-RD); it is associated with increased IgG4 serum concentrations, lymphoplasmacytic infiltrate and phlebitis in the pancreas. Instead, type 2 AIP affects only the pancreas, has normal IgG4 serum concentration and, histologically, shows a neutrophilic inflammation [69]. Hereditary and environmental factors together are thought to induce adaptive immune responses to self-antigens [70]. Microbial antigens may underlie the pathogenesis, but the trigger for the autoimmune cascade remains unknown.

At the beginning of the 2000s, several studies have examined the possible association between *Helicobacter pylori* (*H. pylori*) infection and pancreatic diseases, including AIP [71,72,73,74]. 

In 2010, Jesnowski et al., did not isolate *H. pylori* DNA either from samples or pancreatic juice of patients affected by AIP [75], excluding a direct bacterial infection of the gland. Afterwards, it has been proposed that *H. pylori* could cause AIP due to molecular mimicry mechanism. Guarneri et al. [76], found a significant homology between *H. pylori* a-carbonic anhydrase (a-CA) and human CA type II, enzyme highly expressed in pancreatic ductal cells. Moreover, the homologous CA segments contain the binding motif of the HLA molecule DRB1*0405, which confers a risk for AIP development [76]. In 2009, Frulloni et al. [74] reported that 94% of AIP patients exhibited IgG antibodies to *H. pylori* plasminogen-binding protein (PBP), that is homologous to the human protein ubiquitin-protein ligase E3 component n-recognin 2 (UBR2), highly expressed in pancreatic acinar cells. Instead, this finding was found in only 5% of patients with pancreatic cancer adenocarcinoma (PDAC) and none of patients with alcohol-induced CP or intraductal papillary mucinous neoplasms (IPMN) [74]. Unfortunately, to date, these findings have not been confirmed. At first, Buijs et al. in 2016 [77], and then Detlfsen et al. [78] in 2018, did not find significant differences in serum concentrations of anti-PBP antibodies in AIP patients. Furthermore, in the second study the authors investigated the anti-CAII antibodies serum concentration, but no differences were found with the control group. Finally, in a prospective study, Culver et al. [79], found that the prevalence of gastric ulcerations, exposure to *H. pylori*, cytokine response and immunological memory to *H. pylori* PBP did not differ in IgG4-RD patients compared with control group. Hence, they ruled out a role for *H. pylori* PBP as a microbial antigen in IgG4-RD pathogenesis [79].

Watanabe et al., proved that antigens derived from intestinal microflora, through the activation of NOD-2 and TLR pathways, enhance IgG4 responses by peripheral blood mononuclear cells (PBMCs) from AIP patients, showing a possible involvement of innate immune responses against intestinal microflora in the development of AIP [80]. Several AIP-like experimental models have been induced in transgenic mice inoculating microbial agents: C57BL/6 mice infected with the murine leukemia retrovirus LP-BM5, developed histological findings similar to human AIP [81,82]; in MRL/Mp mice the administration of polyinosinic:polycytidylic acid (poly I:C), a synthetic double-stranded RNA and TLR3 ligand, promotes the development of AIP-like pancreatitis [83,84,85,86]. After showing in their first study that repeated inoculations with heat-killed non-pathogenic *Escherichia coli* (*E. coli*) (ATCC 25922), a common commensal bacterium from gut microbiota, into C57BL/6 mice induced AIP-like pathological alterations accompanied by an elevation in serum IgG [87], Yanagisawa et al. identified [88] the *E. coli* antigen capable of these alterations: FliC, the major component of the flagella, an important cell surface structure of Gram-negative bacteria. In murine model, when injected intraperitoneally, FliC was able to induce AIP-like pancreatitis. Moreover, serum concentration of anti-FliC antibodies was found to be significantly higher in AIP patients than in those with CP, pancreatic cancer, and pancreatic disease-free controls. These findings indicate that FliC from *E. coli* may be involved in the pathogenesis of AIP [88]. Finally, Kamata et al. [89], investigated in a murine model of AIP the role of immune response against intestinal microflora, via plasmacytoid dendritic cells (pDCs) activation, whose products, interferon (IFN)-α and interleukin (IL)-33, are responsible for chronic inflammation and pancreatic fibrosis [89,90,91,92]. They proved that bowel sterilization by broad spectrum antibiotics decreased pancreatic accumulation of pDCs, and consequently IFN-α and interleukin (IL)-33 levels, halting AIP development. They observed a reduction in the microbial diversity as well as alteration in the microbial composition in the feces of mice with AIP, with the abundance of *Bifidobacterium* spp. in the gut of AIP mice. Lastly, they proved that intestinal dysbiosis, due to transmission of intestinal microflora by co-housing or by FMT, makes mice more sensitive to develop experimental AIP when they were injected with a lower dose of an inducing agent [89]. If these data are confirmed in the human form of AIP, patients with this condition might be efficiently treated with a blockade of IFN-α and IL-33, in combination with the normalization of the intestinal microflora.

### 4.4. Microbiota and Type 1 Diabetes Mellitus

Type 1 diabetes (T1D) is an autoimmune disease characterized by the destruction of pancreatic insulin-producing β cells. Susceptibility to T1D is influenced by both genetic and environmental factors that act on immune system dysregulation [93]. The risk of developing T1D, in fact, has been related to viral infections, dietetic factors, and vitamin D deficiency, while the role of antibiotics is still controversial [94].

The impact of intestinal microbiota in T1D predisposition and pathogenesis has been studied in human subjects and in murine models, such as non-obese diabetic mice [95,96]. Early-life events, such as delivery mode, breastfeeding, solid food introduction and antibiotics, strongly influenced the microbiota composition and its interaction with mucosal and systemic immunity [95]. Diabetes resulted to be associated with higher *Bacteroidetes*/*Firmicutes* ratio and lower α-diversity in fecal samples [95,96]. Moreover, subjects affected by T1D showed an increase in levels of *Clostridium*, *Bacteroides dorei* and *vulgatus*, *Blautia*, *Rikenellaceae*, *Ruminococcus* and *Streptococcus*, while *Lactobacillus*, *Bifidobacterium* and SCFAs-producing bacteria were reduced [94,95].

Microbiota dysregulation affects the risk of T1D through different mechanisms, including intestinal permeability [97,98], molecular mimicry [99] and immune system modulation [100,101,102,103].

The intestinal barrier prevents the translocation of toxins, food antigens and infectious factors, which can activate the immune system. Specific microorganisms of gut microbiota, including *Dialister invisus*, *Bifidobacterium longum*, *Gemella sanguinis* and *Clostridium perfringens*, can compromise the intestinal barrier, leading to an increased risk of T1D [94]. Alterations of the barrier function, in fact, are associated with activation and proliferation of islet-autoantigen specific CD8+ T cells, with pro-diabetogenic effect [98].

Moreover, children affected by T1D showed a significantly higher intestinal permeability [98]. 

The molecular mimicry is an important pathogenetic mechanism not only for diabetes but also for other autoimmune diseases. The molecular structure of some bacterial proteins showed to be similar to pancreatic self-antigens. Monofunctional glycosyltransferase protein (MGT) of *Leptotrichia goodfellowii*, for example, showed similarity to islet specific glucose-6-phosphatase related protein and could cause immune cross-reactions [94,104,105].

Innate immunity involves macrophages, granulocytes and natural killer cells, and a key role in the antigen recognition is played by TLRs, which are activated by PAMPs. Many components of intestinal bacteria, such as lipopolysaccharides (LPS), lipoproteins, peptidoglycan and nucleic acids, act like PAMPs and can activate TLRs, with pro-diabetogenic or anti-diabetogenic effects [101]. Different TLRs, in fact, can promote or inhibit autoimmune processes: TLR2-pathway showed to facilitate T1D, while TLR4-signaling resulted to be protective against diabetes development [101]. In addition, some microorganisms can modulate T-cells of B-cells functions. Even if the exact mechanism is still unknown, *Listeria* can induce Th1 response [102], while *Clostridia* can induce regulatory T cells [102]. SCFAs are associated with reduced serum levels of pro-inflammatory cytokines which promote T1D development, such as interleukin-21 (IL-21). They preserve the intestinal barrier function and promote B cells differentiation into plasma-cells and memory cells [103]. Even if there is no current evidence-based treatment for preventing or delaying T1D onset by acting on gut microbiome, the above-mentioned data suggest some possible therapeutic approaches. The immune dysregulation, for instance, can be modulated through the administration of *Escherichia coli Nissle* (*EcN*), which is associated with reduced pathogenic bacteria colonization, decreased serum levels of IL-2 and TNFα and increased IL-10 [95]. Other possible strategies include the correction of dysbiosis and the improvement of microbial diversity through probiotics and prebiotics, such as inulin and oligosaccharides [95]. These treatments have been tested in mouse models, but further studies are needed to prove their efficacy in humans.

### 4.5. Microbiota and Pancreatic Cystic Neoplasms

Pancreatic cystic neoplasms (PCN) represent up to 5% of the total amount of pancreatic cancerous neoplasms [106]. Frequently, the discovery of pancreatic cysts is casual, and these are found during screening or diagnostic procedures for other problems. Diagnostic refinement by computed tomography (CT) and magnetic resonance imaging (MRI) have allowed us to define PCN, undetected in the past. PCN are divided into various types, such as serous cystadenomas (SCA), mucinous cystic neoplasm (MCN) and IPMN; other forms of PCN are very unusual [107]. IPMN can be divided into three types: IPMN main duct, IPMN branch duct and the mixed form [108]. In recent years, researchers tried to find a connection between gut microbiota and cystic pancreatic lesions, but some questions have not been answered yet. Following data reporting the correlation between poor oral health and PDAC [109,110]. Olson et al., in a pilot study, assessed the difference between oral microbiota in patients with PDAC, IPMN and controls [111]. The authors evaluated the characteristics of the oral microbiota of 40 PDAC patients, 39 of whom with IPMN and 58 patients included as control group. Eligible participants were age 21 or over, had not smoked tobacco products in the past year, had not taken antibiotics in the past 30 days, had not been treated for any cancer (other than non-melanoma skin cancer) in the past two years. The 16s rRNA microbiota screening was performed in saliva samples. Patients with PDAC had a higher mean relative proportion of *Firmicutes*/*Bacteroidetes* ratio, while *Proteobacteria* were predominant in controls; in fact, the first group had high levels of *Firmicutes* and *Lactobacillae*. However, differences were observed only in the main relative proportion of some Operational taxonomic unit (OTU). No differences were detected between PDAC and IPMN groups about alpha diversity [111]. Gaiser et al., have reported a prospective study about the harbor of intracystic microbiome in PCN [112]. Patients, undergoing pancreatectomy, were enrolled. Cystic fluid samples were collected at the day of surgery. The authors found that 16s DNA copy number were higher in IPMN with high grade of risk malignancy than benign neoplasms. Moreover, IL-1 beta protein was higher in IPMN and cancer. In particular, *Proteobacteria* were predominant in benign cystic forms, while *Firmicutes* in IPMN with high grade dysplasia (HGD) and PDAC. A sub-analysis was performed to investigate the potential role of invasive endoscopic procedures [i.e., endoscopic ultrasonography-fine needle agoaspiration (EUS-FNA) or endoscopic retrograde cholangiopancreatography (ERCP) or percutaneous transhepatic cholangiography (PTC)] in the microbiota composition. *Fusobacterium nucleatum* (*F. nucleatum*) was one of the predominant pathogens in cystic specimens. qPCR assay confirmed that *F. nucleatum* were increased in IPMN HGD and in PDAC. Previous procedures had no impact on microbiota composition, as well as the previous use of PPI [112]. It is now believed that *F. nucleatum* have an oncogenic role in some cancers, such as colorectal cancer [113] and PDAC itself [114].

Li et al. [115] studied pancreatic cyst fluid (PCF) to determine the genera of bacteria present. Sixty-nine pancreatic fluid samples were collected. Twenty-seven were IPMN, 13 MCN, nine pseudocysts and nine SCA; the rest were classified as “other”. Predominant microorganisms in the cysts were *Bacteroides* spp. and *Escherichia* spp., and even less *Fusobacterium* spp. and *Bacillus* spp. The authors found that PCF contained their bacterial microenvironment with a unique ecosystem. Finally, *Helicobacter* were marginally detected in pancreatic cyst fluid. In this case, more studies are necessary as may be further warranted. 

### 4.6. Microbiota and Pancreatic Ductal Adenocarcinoma

Pancreatic ductal adenocarcinoma is the 12th most common cancer worldwide [116] and the fourth most fatal cancer in both men and women [117]. The incidence of PDAC is higher in North America and in Western Europe and it is more common in men than in women [116].

Risk factors for PDAC include cigarette smoking, obesity and CP. Some chemical compounds, nickel-based, chromium-based, and silica dust have been reported to increase the risk of PDAC too [117]. Moreover, the development of PDAC has been related to dietary factors, *H. pylori* infection, oral, gut and pancreatic microbiota [117]. The mechanisms by which microbiota influences carcinogenesis involve innate and adaptive immune suppression [118,119] and stimulation of pro-carcinogenic cellular pathways [118,120]. In preclinical mice models, in fact, microbial ablation with broad-spectrum antibiotics has been associated with increased presence of intratumoral T cells [117], while the administration of cell-free extracts from fecal samples of gut bacteria from PDAC-bearing mice or cell-free extracts from *Bifidobacterium pseudolongum* were associated with lower expression of MHC II and up-regulation of IL-10 [118]. Moreover, bacteria-associated PAMPs bind to specific TLRs, such as TLR4, thus activating MAP (mitogen-activated protein) Kinase and NF-kB (nuclear-factor kappa-light-chain-enhancer of activated B cells) pathways, which are potent promoter of carcinogenesis [121].

The etiologic relationship between gut microbiota and PDAC has been initially studied in mice models. In these studies, germ-free mice and those treated with oral antibiotics showed a lower incidence of pancreatic cancer, while fecal transplant from PDAC-bearing mice resulted in an increased risk of tumor [118].

In murine models, the presence of PDAC has been related to increased number of gut bacteria from fecal samples belonging to *Bacteroides*, *Firmicutes* and selected genera associated to *Actinobacteria* and *Deferribacteres*. Moreover, different microbiota composition has been related to a different rate of pancreatic carcinoma progression [118]. *Elizabethkingia*, *Enterobacteriaceae* and *Mycoplasmacetaceae* resulted to be associated with slower progression, while aggressive PDAC was associated with *Helicobacteriaceae*, *Bacteroidales* and *Mogibacteriaceae* [118]. In humans, *Proteobacteria*, *Actinobacteria*, *Fusobacteria* and *Verrucomicrobia*, which normally represent a minor proportion of human intestinal microbiome [122], were present in abundance in the gut of PDAC patients if compared to controls [118].

In epidemiological studies, poor oral health and periodontal diseases have been related to PDAC incidence [121]: these data suggest a possible relationship between oral microbiome and pancreatic cancer, as mentioned above. In human studies, in fact, the carriage of *Porphyromonas gengivalis* and *Aggregatibacter actinomycetemcomitans* and a decreased proportion of *Fusobacteria* and its genus *Leptotrichia* resulted to be associated with higher risk of PDAC development [110]. Many studies have demonstrated the role of *Porphyromonas gingivalis*, an oral pathogen that causes periodontitis and gingivitis, in the PDAC. Smoking and alcohol consumption are recognized risk factors for pancreatic cancer, but at the same time they can affect oral microbiome composition. Therefore, the association between oral bacteria and PDAC has been studied in ever and never-smokers and in ever and never-drinkers: the results suggest that the association between oral microbiota and PDAC are not likely due to these potential confounders [110]. The presence and composition of pancreatic microbiota have been associated to PDAC development: cancerous pancreas showed a more abundant microbiota compared to normal pancreas in both murine models and human subjects [118]. Translocation of gut bacteria and their migration into the pancreas have been demonstrated by Pushalkar et al. [118], through comparison between fecal samples and pancreatic microbiota and through oral administration of fluorescently labelled *Enterococcus faecalis* and Green Fluorescent Protein (GFP)-labeled *Escherichia coli*. Thus, gut microbiota could influence pancreatic microenvironment. The composition of pancreatic microbiota in PDAC resulted to be different from normal human pancreas. Thirteen phyla were detected in tumor tissue. The most abundant were *Proteobacteria*, *Bacteroidetes* and *Firmicutes* [118]. Interestingly, *Proteobacteria* only amount for about 8% of gut bacteria from fecal samples in PDAC patients, while they were one of the most represented bacteria in their pancreatic tissue [118]; this suggests a differentially increased translocation of these Gram-negative bacteria to the pancreas. Nowadays, it is believed that *F. nucleatum*, an inhabitant of the oral mucosa, is important in the colon rectal dysplasia and cancer. Similar results were found on PDAC. In a landmark study, Mitsuhashi et al. [114] studied the correlation between PDAC and oral microbiota; they analyzed a database of 283 patients with PDAC who underwent surgery and found *Fusobacterium* spp. in 8.8% of pancreatic cancer samples; the presence of tumor *Fusobacterium* was not associated with any molecular and clinical features, but with worse prognosis. Mitsuhashi et al. suggested that *Fusobacterium* could be a negative prognostic negative independent biomarker for PDAC. 

This data about intestinal, oral and pancreatic microbiota can represent useful screening and prevention tools. In fact, they suggest that the analysis of oral and gut microbiota composition could be used for risk stratification and early diagnosis of PDAC. Moreover, oral antibiotics could be proposed as a preventive therapy for high-risk patients or in association with chemotherapy with a synergic effect [118]. Both these diagnostic and therapeutic implications, however, need further studies to be confirmed.

### 4.7. Microbiota and Pancreatic Neuroendocrine Tumors

Evidences about a link between pancreatic neuroendocrine tumors (pNET) and microbiota are very poor. Currently, therapies for pNET are surgery for not invasive neoplasms, somatostatin analogues (SSA), peptide receptor radionuclide therapy (PRRT), target therapy with Everolimus and Sunitinib and chemotherapy (for advanced NET) [123]. In this scenario, immunotherapy could have a futuristic role; in particular, PD-1 inhibitors and PD-1 ligands are a group of checkpoint inhibitors that have been developed for treatment of some cancers (i.e., melanoma, squamous cell lung cancer, advanced cell renal carcinoma, Merkel-cell carcinoma, a cutaneous neuroendocrine tumor) [124,125]. Specific clinical trials are looking for a role of immune checkpoint inhibitors in pNETs [126], that may conduct to more personalized target therapy; in particular researchers are testing pembrolizumab in NET with advanced stage. In a landmark study of Nghiem et al. [125], they used pembrolizumab as first-line therapy in patients with advanced Merkel-cell carcinoma (a type of skin neuroendocrine tumor) with an objective response rate of 56%.

Recent data support a role for the commensal microbiota in the efficacy of immunotherapy itself. Indeed, gut microbiota may have a “mechanistic impact” on antitumor immunity in the cancer [127]. Matson et al., collected 38 stool samples from metastatic melanoma patients on anti-PD-1 treatment and, after 16S RNA sequencing and quantitative PCR analysis, they identified *Bifidobacterium* spp., *Colinsella aerofaciens*, *Enterococcus faecium* as bacteria associated with beneficial response [127]. They transplanted fecal material from responding patients in a germ-free mice model of melanoma. They observed that FMT could lead to improve tumor control, augmenting T cell responses, and greater efficacy of anti-PD-L1 therapy. 

The commensal microbiota composition might be useful as a biomarker to predict response to checkpoint blockade therapy, and in the future, it could be possible to translate this research for the pNET. However, data are very poor now, and the relationship between immunotherapy and microbiota, in NET patients, is still unclear.

## 5. Conclusions

Although the relationship between microbiota and pancreatic diseases is an innovative and inspiring field of research, several points warrant further clarification. At present, the association between pancreatic diseases and microbiota is not well established. Hence, the correlation between microbiota and pancreas remains fraught with challenges. Given the potential role of microbiota in the pancreatic diseases, technological improvement in microbiota manipulation as well as randomized controlled trials could represent a powerful therapeutic strategy for their prevention and treatment.

## Figures and Tables

**Table 1 jcm-09-03535-t001:** Microbial changes in the pancreatic diseases.

Author, Year, [Ref.]	Study Population	Material	Disease	Microbial Changes
Tan, 2015, [47]	Humans	Fecal samples	SAP and MAP	↑*Enterococci* and ↓*Bifidobacteria* in patients with SAP versus patients with MAP
Zhang, 2018, [49]	Humans	Fecal samples	AP	↑*Proteobcateria* and *Bacteroidetes* ↓*Actinobacteria* and *Firmicutes*
Capurso, 2016, [58]	Humans	Breath tests	CP	1/3 of CP patients show SIBO
Nishiyama, 2018, [60]	Mouse	Cecum, transverse colon and fecal samples	CP	↑*Akkermansia muciniphila* and ↑*Lactobacillus reuteri* in mice treated with PERT
Li, 2017, [115]	Humans	Pancreatic cystic fluid	PCN	↑↑*Bacteroides* spp. and *Escherichia/Shigella* spp. + ↑*Fusobacterium* spp. and *Bacillus* spp. in PCN *Helicobacter* were marginally detected in pancreatic cyst fluid
Knip, 2017, [95]	Humans	Fecal samples	T1D	↑*Bacteroidetes*/*Firmicutes* ratio + ↑*Clostridium*, *Bacteroides dorei* and *vulgatus*, *Blautia*, *Rikenellaceae*, *Ruminococcus* and *Streptococcus* ↓*Lactobacillus*, *Bifidobacterium* and short-chain fatty acids-producing bacteria
Pushalkar, 2018, [118]	Humans	PDAC specimens and fecal samples	PDAC	↑*Proteobacteria*, *Actinobacteria*, *Fusobacteria* and *Verrucomicrobia* in PDAC
Fan, 2018, [110]	Humans	Oral samples	PDAC	↑*Porphyromonas gengivalis* and *Aggregatibacter actinomycetemcomitans* and ↓*Fusobacteria* and its genus *Leptotrichia*: high risk of PDAC development
Mitsuhashi, 2015, [114]	Humans	PDAC specimens	PDAC	*Fusobacterium* spp. observed in 8.8% of PDAC specimens
Olson, 2017, [111]	Humans	Oral samples	PDAC e IPMN	↑*Firmicutes*/*Bacteroidetes* ratio + ↑*Lactobacillae* in PDAC and IPMN (mean relative proportion) ↑*Proteobacteria* in healthy controls (mean relative proportion)
Gaiser, 2019, [112]	Humans	Cystic fluid samples from resected pancreas	PDAC and PCN	↑*Firmicutes* in IPMN-HGD and PDAC ↑*Fusobacterium nucleatum* in IPMN-HGD and PDAC ↑*Proteobacteria* in benign cystic neoplasms

AP: acute pancreatitis; CP: chronic pancreatitis; IPMN-HGD: intraductal papillary mucinous neoplasm with high grade dysplasia; LP: Lactobacillus plantarum; MAP: moderate acute pancreatitis; PCN: pancreatic cystic neoplasms; PDAC: pancreatic ductal adenocarcinoma; PERT: pancreatic enzyme replacement therapy; SAP: severe acute pancreatitis; SIBO: small intestine bacterial overgrowth; T1D: type 1 diabetes.

**Table 2 jcm-09-03535-t002:** Randomized clinical trial about the use of probiotics, prebiotics or synbiotics in acute and chronic pancreatitis.

Author, Year, [Ref]	Type of Study	Disease	Patients	Results	Conclusions
1. Olah, 2002, [50]	Randomized clinical trial	Acute pancreatitis (AP)	45 patients with AP: 22 patients with live *Lactobacillus plantarum* (*LP*) 299 for 1 week by nasojejunal tube 23 with heat-killed *LP299*	Infected pancreatic necrosis in 1/22 patient in the treatment group vs. 7/23 in the control group (*p* = 0.023)	Number of surgical treatments and pancreatic sepsis could be reduced by Supplementary live *LP 299*
2. Olah, 2007, [51]	Prospective, randomized, double-blind study	Severe acute pancreatitis (SAP)	62 patients with SAP: 33 patients received four different Lactobacilli preparations + prebiotics by nasojejunal feeding 29 patients received only prebiotics	SIRS and MOF in 8/33 in the first group vs. 14/29 in the second group (*p* < 0.05) Total complications were higher in the second group compared to the first group (*p* <0.05) Lower rate of organ failure in the first (3.0%) vs. the control group (17.2%)	Early nasojejunal feeding with synbiotics could prevent organ dysfunctions in SAP
3. Qin, 2008, [48]	Prospective, randomized, single-blinded study	AP	74 patients with AP: 36 patients treated with LP enteral feeding (*n* = 36) 38 patients treated with parenteral nutrition (PN) group	38.9% patients in enteral feeding group were colonized with multiple organisms vs. 73.7% in the PN group (*p* < 0.01) 30.6% patients in the enteral feeding group were colonized with pathogenic organisms vs. 50% patients in PN group (*p* < 0.05)	Disease severity could be reduced by enteral feeding with LP with better clinical outcomes
4. Besselink MG, 2008, [53]	Multicenter, randomized, double-blind, placebo-controlled trial	SAP	296 patients with predicted SAP: 152 patients treated with a multispecies probiotic formulation 144 patients with placebo	Infectious complications in 46/152 patients (30%) in probiotic group vs. 41/144 (28%) in placebo group (RR 1.06; 95% CI 0.75–1.51) 24/152 patients (16%) in first group died versus 9/144 (6%) in placebo group (RR 2.53; 95% CI 1.22–5.25)	Probiotic did not reduce the risk of infectious complications in SAP. Even, probiotics were associated with an increased risk of mortality
5. Gou, 2014, [55]	Systematic review and meta-analysis of randomized controlled trials	SAP	6 trials with an aggregate total of 536 patients	Probiotics did not impact the pancreatic infection rate (RR = 1.19, 95% CI 0.74–1.93; *p* = 0.47), total infections (RR = 1.09; 95% CI 0.80–1.48; *p* = 0.57), operation rate (RR = 1.42, 95% CI = 0.43 to 3.47; *p* = 0.71) and mortality (RR 0.72; 95% CI = 0.42–1.45; *p* = 0.25).	Clinical outcomes of patients with SAP were not modified by probiotics
6. dos Santos, 2017, [65]	Prospective, randomized, controlled, double blind trial	Chronic pancreatitis (CP)	60 patients with chronic pancreatitis: synbiotics administered to the intervention group 12 g/day of maltodextrin (medium absorption complex carbohydrate) to the control group	Important reduction of bowel frequency in treatment group: Average bowel frequency before treatment: 2.33 (*p* < 0.153) 2nd month of treatment: 1.47 (*p* = 0.002) 3rd month: 1.37 (*p* = 0.012) No change in bowel frequency in the control group (*p* = 0.157)	Clinical outcomes of patients with CP could be ameliorated by synbiotics

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
