# Peer review of "Pancreatic Diseases and Microbiota: A Literature Review and Future Perspectives"

_jcm, 2020, doi:10.3390/jcm9113535_

Round 1

Reviewer 1 Report

In this review article the authors have taken on a large topic but have admirably focused on the key literature to summarize what is known about the microbiota and various pancreatic disease states. Their revisions address my scientific concerns.

Reviewer 2 Report

All my suggestions have been corrected. I think the review is adequate to accept.

Reviewer 3 Report

The authors provide a comprehensive review of the data regarding the role of the gut microbiome in pancreatic diseases. This is an area about which little is known, and their article provides a useful new resource. Several areas require clarification:

-Page 2, line 1: Would rephrase this sentence; stating that commensals are "higher than" pathologic species is unclear.

-Page 2, paragraph 3: Please elaborate on explanation of alpha and beta diversity. (Ex. alpha diversity does not only reflect "amount," but also evenness; beta diversity warrants more detailed summary).

-Page 3, line 137-138: The statement that SIBO and bacterial translocation explain the majority of complications of acute pancreatitis is too strongly worded based on the evidence provided here. In general, the evidence for gut microbiota playing a significant role in acute pancreatitis seems limited, and this section of the manuscript can be rephrased to reflect this uncertainty.

-Multiple sections in the manuscript refer to "gut bacteria" (as one example-- Page 4, paragraph 3). Please be specific each time: which samples? Duodenal, fecal, other? One can imagine very different mechanisms if the duodenal microbiota are being considered as opposed to fecal microbiota.

-Overall, the data are the most compelling for the role of gut microbiota in chronic pancreatitis, PDAC, and possibly pancreatic cyst neoplasms. 

Author Response

This manuscript is a resubmission of an earlier submission. The following is a list of the peer review reports and author responses from that submission.

Round 1

Reviewer 1 Report

This manuscript attempts to review the convergence of two fields, that of pancreatic diseases and of the microbiome. This is a challenging task given the largely pre-clinical nature of the microbiome field.

Perhaps the most succinct and useful statement is that of the authors when they state that "data about correlation between microbiota and pancreatic diseases are still scarce and controversial"

The manuscript has occasional awkward (i.e. line 44), confusing (i.e. "less multiple pathogenic organisms" (line 136), and incorrect (i.e. "intestinal bowel disease" (line 13, I suspect that inflammatory bowel disease is intended) use of English. Other examples of problematic writing is: "in addiction to standard therapies" (line 20) and the repeated misspelling of synbiotics as symbiotics.

Ideally a review article helps the reader by making the primary literature more understandable. At times this manuscript fails to do this. One example is the following sentence: "they suggested to use a higher dose of probiotics, but the probiotics should be stopped as soon as possible to prevent bacterial overgrowth". I do not know how to interpret this: should they be used or stopped?

It is not clear why the authors state that "nowadays the real mechanism of PERT is unknown..." (line 189). Do they not function as digestive enzymes in patients with pancreatic exocrine insufficiency? In patients with maldigestion, PERT could help to normalize digestion and thereby limit nutrients that would otherwise be available to the microbiome. Or perhaps these enzymes could have direct effects on microbes. Such ideas could be explored/discussed in the manuscript.

Regarding the organization of the review, it may make more sense to place the section on pancreatic cancer immediately after the section on pancreatic cysts (rather than place the diabetes section in between the two).

Finally, a major challenge regarding the literature at the intersection of pancreatic disease and the microbiome is sampling. Many studies analyze fecal samples which are collected at distance from the pancreas. It may be useful for the authors to include discussion on this challenge as it relates to the field.

Reviewer 2 Report

The review by Gesualdo et al, though very readable, has numerous issues. These include bias, over emphasis of causality while the data is merely associational, and contextual errors. The whole paper is full of such problems. These make the study misleading for the reader who is new to the field. I will emphasize some of the ones I noted under “weaknesses”, though there are several more examples all through the manuscript.

Strengths: The descriptions under the various heading are good, giving outlines of what is normal, and explaining the terminology used in microbiome studies.

Weaknesses:

The methods are non-systematic, and merely descriptive. This leads to bias in the text that will influence the reader. For example, there are more than 20 papers on the microbiome in pancreatitis alone. The choice of studies quoted seem to be biased, with a goal to present the authors’ hypothesis rather than being a fair representation of what majority of the studies showed.

There is no critique or distinction provided for studies that attempted to change the mirobiome (e.g those by Olah) and their clinical outcomes, vs. whether changes in the microbiome were actually verified such as with a 16S sequencing analysis. This is important since an intervention may affect outcomes irrespective of leaky gut (which is over emphasized, e.g. in page 3 lines 120-140). Other factors include changes in gut motility, microbial products that neutralize or bind (toxins or fatty acids) those produced by other bacteria etc.

There are not comments on alpha or beta microbiome diversity in pancreatic diseases.

There is an inappropriate and excessive depiction of “causality” when discussing findings, while in reality the finding was merely an association noted in human disease. For example when describing dysbiosis, (page 2 first paragraph); the dysbiotic environment is linked to tight junction reduction, compromised mucosal integrity, and bacterial translocation. This is incorrect, since dysbiosis (e.g. from motility disorders, blind loop syndrome, antibiotic use) in most cases does not lead to these problems. Moreover, severe compromise of mucosal integrity (e.g. ulcerative colitis, ischemic colitis) does not lead to clinically relevant bacteremia or septicemia in most cases.

Similarly, in page 3, line 105-108, the microbiome changes are attribute to pancreatic juice changes after a pancreatico-duodenectomy, while ignoring the anatomical alterations, such as the stomach (and therefore acid) and anastomosis with the liver/ bile being downstream to the pancreas after surgery.

Irrelevant experimental findings are over-emphasized (e.g. genetic deletion of orai1, on page), while ignoring that severe exocrine pancreatic insufficiency, while changing the microbiome, is not lethal in most scenarios, unless associated with severe diabetes, nutritional deficiencies or pancreatic cancer.

There are several typographical and contextual errors. For e.g. Line 20 in abstract uses “addiction”, which should have been addition. Line 30 of introduction uses helminths as an example of microbiota, which is incorrect. Line 115 page 3 mention antibiotics only for pancreatic necrosis while ignoring their common use, when coexistent cholangitis is suspected in biliary pancreatitis. Intestinal ischemia (line 124, page 3) is blamed for bacterial translocation and MODS, however MODS is rare (<5%) with definite perforations and translocations such as diverticulitis, fistulizing Crohn’s and ischemic colitis, unless severe.

Reviewer 3 Report

he paper by Gesualdo describes efficacy of microbiota manipulation in pancreatic diseases. The authors showed review current research and therapy as well as suggest prospectively diagnosis and therapy using microbiota in pancreatic diseases. The authors have some great review but some of description need to be clarified.

  1. Please clarify both intestinal on line 13 in abstract. is it loss of words?
  2. You need reference about helminths in microbiota (line 29 in introduction)
  3. Line 114-115 is unclear. What is cardiac perfusion reference?
  4. Please clarify pro-diabetogenic or anti-diabetogenic effect. (section 4.4)
  5. Last 6 sentences don’t refer type 1 diabetes particularly reference 99 is incorrect.